# An enhanced pairing-free certificateless directed signature scheme

**Kaiqin Yang** [ORCID] *

School of Control Science and Engineering, Shandong University, Jinan, Shandong, China

* yangkaiqin2021@163.com

## Abstract

Directed signature is a special cryptographic technique in which only the verifier designated by the signer can verify the validity of the signature. Directed signature can effectively protect the privacy of the signer's identity, so it is very suitable for medical records, taxation, and other fields. To improve the security and performance of the directed signature scheme, Gayathri et al. proposed the first certificateless directed signature (CLDS) scheme without bilinear pairing and claimed that their CLDS scheme could withstand Type I and Type II attacks. In this article, we provide two attack methods to assess the security of their CLDS scheme. Unfortunately, our results indicate that their CLDS scheme is insecure against Type I and Type II attacks. That is, their CLDS scheme does not meet the unforgeability and cannot achieve the expected security goals. To resist these attacks, we present an improved CLDS scheme and give the security proof. Compared with similar schemes, our scheme has better performance and higher security.

**Data Availability Statement:** All relevant data are within the paper and its Supporting information files.

**Funding:** The author(s) received no specific funding for this work.

## 1 Introduction

Digital signature is one of the most important technologies for ensuring data security in insecure communication networks [1–3]. It can solve the problems of forgery, denial, impersonation and tampering. Digital signature provides trusted identity authentication [4] and data exchange services [5], so it is widely employed in smart city [6], smart transportation [7, 8], smart medicine [9, 10], smart grid [11] and other fields [12]. In traditional digital signature schemes, anyone can verify the signature's validity by using the signer's public key. However, in some practical circumstances (such as hospitals, shopping malls), the user's public key is the private information that symbolizes the user's identity. For example, patients do not want their health information shared with anybody other than their doctors, and consumers do not want their shopping information shared with anyone else. This requires that the user's signature does not support public verification [13–15].

To accomplish these security functionalities, directed signature is introduced as a new signature technology [16]. Directed signature allows the signer to control the verifier of his or her signature. On the one hand, only the designated verifier can decide whether the signature is legitimate, while others can only check the signature's validity with the signer's or designated

**Competing interests:** The authors have declared that no competing interests exist.

verifier's help. On the other hand, when a disputed message occurs, the legitimacy of the signature can be proved to the third party by the signer and the designated verifier. On the basis of ordinary signature, directed signature adds the feature of safeguarding the signer's identity privacy, preventing any third party from directly verifying the signature using the signer's public key. As a result, directed signature is well suited to real scenarios in which the signer's identity anonymity is required [17–19].

Since Lim and Lee [16] introduced the concept of directed signature, some directed signature schemes [20–22] based on the framework of public key infrastructure have been presented. However, these schemes require huge computational and communication overhead to manage certificates. Identity-based directed signature [23–25] does not require a certificate to authenticate the user's public key, but a trusted key generation center (KGC) is required to produce a private key for the user. As a result, the key escrow problem exists in the identity-based directed signature scheme. In other words, KGC has the ability to sign any message on behalf of the user. To address these issues, certificateless directed signature (CLDS) is presented [26]. In CLDS, the user's private key is created independently by a semi-trusted KGC and the user, so that KGC cannot obtain the user's final private key. CLDS combines the benefits of both certificateless signature [27] and directed signature, making it more suitable for insecure communication network environments [28].

The first CLDS scheme was presented by Wan [26], but their scheme has poor computational performance because of the time-consuming bilinear pairing operation. To achieve high efficiency, Gayathri et al. [29] constructed a pairing-free CLDS scheme in 2021 and proved that their CLDS scheme is unforgeable against Type I and Type II attacks in the random model. However, in this article, we give two attack methods to prove that their CLDS scheme is insecure against Type I and Type II attackers. Specifically, both dishonest users and malicious KGC have the capacity to forge the legal signature of any message. This demonstrates that their CLDS scheme is riddled with security issues.

## 1.1 Our contribution

The following is a brief summary of our work.

1. We show that the CLDS scheme proposed by Gayathri et al. [29] cannot withstand Type I attackers by using an attack approach.

2. We show that Gayathri et al.'s CLDS scheme [29] is vulnerable to Type II attackers by employing another attack approach.

3. To improve security, we provide an enhanced CLDS scheme.

4. We give the security proof of the modified scheme. Furthermore, according to the results of the analysis, our scheme outperforms similar schemes in terms of performance and security.

## 1.2 Organization

The remainder of this article is structured in the following manner. Section 2 introduces some necessary preliminaries. Section 3 describes Gayathri et al.'s CLDS scheme [29] and analyzes its security in Section 4. Section 5 presents our improved CLDS scheme, as well as its security proof and performance analysis. Finally, the article's conclusions are presented in Section 6.

## 2 Preliminaries

### 2.1 Elliptic curve group

Assume that $F_p$ is a finite field. The equation $y^2 = x^3 + ax + b$ defines an elliptic curve $E$ on $F_p$ that meets the condition $4a^3 + 27b^2 \neq 0$, where $a, b \in F_p$. Let $O$ be an infinite point of $E$. $G = \{(x, y) \in E: x, y \in F_p\} \cup \{O\}$ is an additive elliptic curve cyclic group [30].

The security of Gayathri et al.'s CLDS scheme [29] is based on the following elliptic curve discrete logarithm problem (ECDLP).

Given $xP \in G$, the ECDLP is to calculate $x \in Z_q^*$, where $P$ is a generator of $G$.

### 2.2 Definition and security model of the CLDS scheme

As defined in [29], a CLDS scheme is composed of the following six algorithms.

1. **Setup**: On input a security parameter $\eta$, this algorithm is executed by KGC and outputs the master secret key and public parameters *params*.

2. **Partial Secret Key Gen**: On input an identity $ID_i$, this algorithm is executed by KGC and outputs a partial private key $D_i$.

3. **User Key Gen**: The user runs this algorithm to generate its public key $PK_i$ and secret key $SK_i$.

4. **Signature Generation**: Given a message $m$ and a public key $PK_V$ of a designated verifier, a signer uses its secret key $SK_S$ and public key $PK_S$ to generate a signature $\sigma$ of $m$.

5. **Direct Verification**: On input a signature $\sigma$ of $m$ and the signer's public key-identity pair $(PK_S, ID_S)$, the designated verifier uses its secret key $SK_V$ to check the validity of the signature. This algorithm outputs *accept* if $\sigma$ is legal; otherwise, it outputs *reject*.

6. **Public Verification**: Given a signature $\sigma$ of $m$, a value $Aid$ calculated by the signer or the designated verifier, the signer's public key-identity pair $(PK_S, ID_S)$ and the designated verifier's public key-identity pair $(PK_V, ID_V)$, this algorithm outputs *accept* if $\sigma$ is legal; otherwise, it outputs *reject*.

As discussed in [26, 29], a secure CLDS scheme must resist the following two types of attackers.

1. *Type I Attacker* $\mathcal{A}_1$: It is an external adversary who can launch the public key replacement attack on any user. However, $\mathcal{A}_1$ cannot obtain KGC's master secret key.

2. *Type II Attacker* $\mathcal{A}_2$: It is a malicious KGC that can create the partial private key of any user and modify the values of system parameters, but cannot replace the user's public key and obtain the user's secret key.

To describe the unforgeability of a CLDS scheme, the following oracles are defined by security games between the challenger and the adversary.

- **Reveal Partial Secret Key Oracle** $\mathcal{O}_{PSK}$: On receiving an identity $ID_i$ from the adversary, the challenger obtains $D_i$ by running the **Partial Secret Key Gen** algorithm and returns it to the adversary.

- **Create User Oracle** $\mathcal{O}_{CU}$: When receiving an identity $ID_i$ from the adversary, the challenger obtains a public key $PK_i$ by running the **User Key Gen** algorithm and returns it to the adversary.

- **Reveal Secret Key Oracle** $\mathcal{O}_{SK}$: On receiving an identity $ID_i$ from the adversary, the challenger obtains a secret key $SK_i$ by running the **User Key Gen** algorithm and returns it to the adversary.

- **Replace Public Key Oracle** $\mathcal{O}_{RK}$: On receiving a tuple $(ID_i, x_i^*, PK_i^*)$ from from the adversary, the challenger replaces $(x_i, PK_i)$ of $ID_i$ with $(x_i^*, PK_i^*)$, respectively.

- **Sign Oracle** $\mathcal{O}_S$: When receiving two identities $(ID_S, ID_V)$ and a message $m$ from the adversary, the challenger obtains a signature $\sigma$ of $m$ by running the **Signature Generation** algorithm and returns it to the adversary.

- **D. Verify Oracle** $\mathcal{O}_{DV}$: When receiving two identities $(ID_S, ID_V)$, a message $m$ and a signature $\sigma$ from the adversary, the challenger runs the **Direct Verification** algorithm and returns the corresponding output to the adversary.

- **P. Verify Oracle** $\mathcal{O}_{PV}$: When receiving two identities $(ID_S, ID_V)$, a message $m$ and a signature $\sigma$ from the adversary, the challenger runs the **Public Verification** algorithm and returns the corresponding output to the adversary.

  **Game 1**: The adversary in this game is a Type I attacker $\mathcal{A}_1$.

- **Initialization Phase**: The challenger runs the **Setup** algorithm and sends *params* to $\mathcal{A}_1$.

- **Queries Phase**: $\mathcal{A}_1$ is allowed to adaptively query seven oracles $\mathcal{O}_{PSK}$, $\mathcal{O}_{CU}$, $\mathcal{O}_{SK}$, $\mathcal{O}_{RK}$, $\mathcal{O}_S$, $\mathcal{O}_{DV}$ and $\mathcal{O}_{PV}$.

- **Forgery Phase**: Eventually, $\mathcal{A}_1$ outputs a tuple $(ID_S^*, ID_V^*, m^*, \sigma^*)$. $\mathcal{A}_1$ wins in this game if $\sigma^*$ is a valid signature on $m^*$ under $ID_S^*$ and $ID_V^*$, and the following conditions hold.

  1. The oracle $\mathcal{O}_{PSK}$ has never been involved for $ID_S^*$.

  2. The oracle $\mathcal{O}_S$ has never been involved for $(ID_S^*, ID_V^*, m^*)$.

  **Game 2**: The adversary in this game is a Type II attacker $\mathcal{A}_2$.

- **Initialization Phase**: $\mathcal{A}_2$ runs the **Setup** algorithm, and sends the master secret key and *params* to the challenger.

- **Queries Phase**: $\mathcal{A}_2$ is allowed to adaptively query the oracles $\mathcal{O}_{CU}$, $\mathcal{O}_{SK}$, $\mathcal{O}_S$, $\mathcal{O}_{DV}$ and $\mathcal{O}_{PV}$.

- **Forgery Phase**: Eventually, $\mathcal{A}_2$ outputs a tuple $(ID_S^*, ID_V^*, m^*, \sigma^*)$. $\mathcal{A}_2$ wins in this game if $\sigma^*$ is a valid signature on $m^*$ under $ID_S^*$ and $ID_V^*$, and the following conditions hold.

  1. The oracle $\mathcal{O}_{SK}$ has never been involved for $ID_S^*$.

  2. The oracle $\mathcal{O}_S$ has never been involved for $(ID_S^*, ID_V^*, m^*)$.

  **Definition 1**. A CLDS scheme is said to be existentially unforgeable, if there is no polynomial time attacker ($\mathcal{A}_1$ and $\mathcal{A}_2$) can win in the above two games with a non-negligible probability.

## 3 Review of Gayathri et al.'s CLDS scheme

In this section, Gayathri et al.'s CLDS scheme [29] is briefly described as follows.

1. **Setup**: KGC selects an elliptic curve group $G$ of prime order $q$ and a generator $P$. Then, KGC randomly selects $s \in Z_q^*$, calculates $P_{pub} = sP$, and sets the master secret key as $msk = s$.

Next, KGC selects three hash functions $H_1, H_2, H_3 : \{0, 1\}^* \rightarrow Z_q^*$. Finally, KGC publicly broadcasts the parameters $params = \{G, q, P, H_1, H_2, H_3, P_{pub}\}$.

2. **Partial Secret Key Gen**: Given the identity $ID_i$ of a user, KGC does as follows.

   a. Select $r_i \in Z_q^*$ at random and calculate $R_i = r_i P$.

   b. Calculate $h_{1i} = H_1(ID_i, R_i, P_{pub})$.

   c. Calculate $d_i = r_i + h_{1i} s \bmod q$.

   d. Send the partial private key $D_i = (R_i, d_i)$ to the user secretly.

   e. After receiving $D_i$ from KGC, the user checks the validity of its partial private key through the following equation.

   $$d_i P = R_i + h_{1i} P_{pub}$$

3. **User Key Gen**: A user with identity $ID_i$ does as follows.

   a. Select $x_i \in Z_q^*$ randomly and calculate $X_i = x_i P$.

   b. Set its secret key as $SK_i = (x_i, d_i)$.

   c. Set its public key as $PK_i = (X_i, R_i)$.

4. **Signature Generation**: Given the public key $PK_V$ and identity $ID_V$ of a designated verifier, a signer with identity $ID_S$ performs the following operations to sign a message $m$.

   a. Select $t_1, t_2 \in Z_q^*$ randomly, and calculate $U_S = t_1 P$ and $V_S = t_2 P$.

   b. Calculate $W_S = U_S + t_2 X_V$.

   c. Calculate $h_2 = H_2(m, ID_S, ID_V, U_S, R_S)$ and $h_3 = H_3(m, ID_S, ID_V, U_S, R_S, h_2)$ by using its identity $ID_S$ and public key $PK_S = (X_S, R_S)$.

   d. Calculate $k_S = h_2 d_S + h_3 t_2 + h_2 x_S \bmod q$ by using its secret key $SK_S = (x_S, d_S)$.

   e. Set $\sigma_S = (k_S, W_S, V_S)$ as the signature of $m$.

5. **Direct Verification**: After receiving a signature $\sigma_S = (k_S, W_S, V_S)$ on a message $m$ from a signer with identity $ID_S$, a designated verifier with identity $ID_V$ performs the following operations to verify the validity of the signature.

   a. Calculate $Y_V = W_S - x_V V_S = U_S$ by using its secret key $SK_V = (x_V, d_V)$.

   b. Calculate $h_2 = H_2(m, ID_S, ID_V, Y_V, R_S)$.

   c. Calculate $h_3 = H_3(m, ID_S, ID_V, Y_V, R_S, h_2)$.

   d. Verify the following signature verification equation.

   $$(k_S P - h_2(R_S + h_{1s} P_{pub} + X_S))h_3^{-1} = V_S$$

   If it holds, $\sigma_S$ is valid and accepted by the verifier; else, the verifier rejects $\sigma_S$.

6. **Public Verification**: Given a signature $\sigma_S = (k_S, W_S, V_S)$ on a message $m$ under two identities $ID_S$ and $ID_V$, any third party does as follows.

a. Obtain the value $Aid = U_S = Y_V$ calculated by the signer with $ID_S$ or the verifier with $ID_V$.

b. Calculate $h_2 = H_2(m, ID_S, ID_V, Aid, R_S)$.

c. Calculate $h_3 = H_3(m, ID_S, ID_V, Aid, R_S, h_2)$.

d. Verify the following signature verification equation.

$$(k_S P - h_2(R_S + h_{1S}P_{pub} + X_S))h_3^{-1} = V_S$$

If it holds, $\sigma_S$ is valid; else, $\sigma_S$ is invalid.

## 4 Security analysis of Gayathri et al.'s CLDS scheme

In [29], Gayathri et al. claimed that their CLDS scheme could withstand Type I and Type II attackers. In this section, we give two concrete attack methods to prove that their CLDS scheme is insecure against Type I and Type II attacks. As a result, their CLDS scheme has serious security weaknesses.

### 4.1 Type I attack

A Type I attacker $\mathcal{A}_1$ selects a target user whose identity and public key are $ID_S$ and $PK_S = (X_S, R_S)$, respectively. $\mathcal{A}_1$ can forge the valid signature of any selected message by replacing the public key of the target user. Let the identity and public key of the designated verifier be $ID_V$ and $PK_V = (X_V, R_V)$, respectively. $\mathcal{A}_1$ initiates the following attack operations.

1. Calculate $h_{1S} = H_1(ID_S, R_S, P_{pub})$.

2. Randomly select $y \in Z_q^*$ and calculate

$$X_S^* = yP - R_S - h_{1S}P_{pub}.$$

3. Replace the previous public key $PK_S$ of the target user with the new public key
   $PK_S^* = (X_S^*, R_S)$.

4. Randomly select $t_1^*, t_2^* \in Z_q^*$, and calculate $U_S^* = t_1^*P$, $V_S^* = t_2^*P$ and $W_S^* = U_S^* + t_2^*X_V$.

5. Select a message $m^*$, and calculate $h_2^* = H_2(m^*, ID_S, ID_V, U_S^*, R_S)$ and
   $h_3^* = H_3(m^*, ID_S, ID_V, U_S^*, R_S, h_2^*)$.

6. Calculate $k_S^* = h_2^*y + h_3^*t_2^* \bmod q$.

7. Output $\sigma_S^* = (k_S^*, W_S^*, V_S^*)$ as a forged signature of $m^*$.

The following equation shows that the signature $\sigma_S^* = (k_S^*, W_S^*, V_S^*)$ forged by $\mathcal{A}_1$ is considered valid and accepted by the verifier with identity $ID_V$, because

$$
\begin{aligned}
k_S^*P &= (h_2^*y + h_3^*t_2^*)P \\
&= h_2^*(yP) + h_3(t_2^*P) \\
&= h_2^*(X_S^* + R_S + h_{1S}P_{pub}) + h_3V_S^*.
\end{aligned}
$$

Then, we have

$$
\begin{aligned}
& (k_S^* P - h_2^*(R_S + h_{1S}P_{pub} + X_S^*))(h_3^*)^{-1} \\
& = (h_2^*(X_S^* + R_S + h_{1S}P_{pub}) + h_3 V_S^* - h_2^*(R_S + h_{1S}P_{pub} + X_S^*))(h_3^*)^{-1} \\
& = h_3^* V_S^*(h_3^*)^{-1} \\
& = V_S^*.
\end{aligned}
$$

Therefore, the forged signature $\sigma_S^*$ meets the signature verification equation in the **Direct Verification** algorithm. During the above-mentioned attack, $\mathcal{A}_1$ does not gain any knowledge about the target user's partial private key. That is, $\mathcal{A}_1$'s attack against the CLDS scheme of Gayathri et al. [29] is successful. As a result, Gayathri et al.'s CLDS scheme [29] is insecure against Type I attacks.

## 4.2 Type II attack

Assume that $ID_S$ and $PK_S = (X_S, R_S)$ are the identity and public key of the target user attacked by a Type II attacker $\mathcal{A}_2$, respectively. Let the identity and public key of the designated verifier be $ID_V$ and $PK_V = (X_V, R_V)$, respectively. $\mathcal{A}_2$ knows the master secret key $s$, so $\mathcal{A}_2$ can calculate the target user's partial private key and modify system parameters. $\mathcal{A}_2$ performs the following attack operations to forge the valid signature of any message.

1. Randomly select $z \in Z_q^*$ and calculate

$$
R_S' = zP - X_S.
$$

2. Replace the previous value $R_S$ with the new value $R_S'$.

3. Calculate $h_{1S}' = H_1(ID_S, R_S', P_{pub})$.

4. Select $t_1', t_2' \in Z_q^*$ randomly, and calculate $U_S' = t_1'P$, $V_S' = t_2'P$ and $W_S' = U_S' + t_2'X_V$.

5. Select a message $m'$, and calculate $h_2' = H_2(m', ID_S, ID_V, U_S', R_S')$ and $h_3' = H_3(m', ID_S, ID_V, U_S', R_S', h_2')$.

6. Calculate $k_S' = h_2'(z + sh_{1S}') + h_3't_2' \bmod q$.

7. Output $\sigma_S' = (k_S', W_S', V_S')$ as a forged signature of $m'$.

The signature $\sigma_S' = (k_S', W_S', V_S')$ forged by $\mathcal{A}_2$ is considered legal and accepted by the verifier whose identity is $ID_V$, because

$$
\begin{aligned}
k_S' P & = (h_2'(z + sh_{1S}') + h_3't_2')P \\
& = h_2'(zP + h_{1S}'(sP)) + h_3'(t_2'P) \\
& = h_2'((R_S' + X_S) + h_{1S}'P_{pub}) + h_3'V_S' \\
& = h_2'(R_S' + h_{1S}'P_{pub} + X_S) + h_3'V_S'
\end{aligned}
$$

It is easy to derive the following equation from the above equation.

$$
\begin{aligned}
& (k'_S P - h'_2(R'_S + h'_{1S}P_{pub} + X_S))(h'_3)^{-1} \\
&= (h'_2(R'_S + h'_{1S}P_{pub} + X_S) + h'_3 V'_S - h'_2(R'_S + h'_{1S}P_{pub} + X_S))(h'_3)^{-1} \\
&= h'_3 V'_S (h'_3)^{-1} \\
&= V'_S.
\end{aligned}
$$

This demonstrate that the forged signature $\sigma'_S$ satisfies the signature verification equation in the **Direct Verification** algorithm. In the above attack, the secret key $x_S$ of the target user is unknown to $\mathcal{A}_2$. Therefore, $\mathcal{A}_2$'s forgery attack against the CLDS scheme of Gayathri et al. [29] is successful. In other words, Gayathri et al.'s CLDS scheme [29] is also insecure against Type II attacks.

## 5 Our improved CLDS scheme

The reason why Gayathri et al.'s CLDS scheme [29] cannot resist the public key replacement attack is that $\mathcal{A}_1$ does not require the target user's partial private key for forging the signature. Furthermore, the reason why Gayathri et al.'s CLDS scheme [29] cannot withstand Type II attacks is that $\mathcal{A}_2$ avoids the target user's secret key by changing the value $R_S$.

### 5.1 Our construction

To resist the two types of forgery attacks, we modify the CLDS scheme of Gayathri et al. [29] as follows.

1. The **Setup** algorithm is the same as the corresponding algorithm in the original scheme, and the only difference is to add a hash function $H_4 : \{0,1\}^* \rightarrow Z_q^*$.

2. The algorithms **Partial Secret Key Gen** and **User Key Gen** are the same as the corresponding algorithms in the original scheme.

3. In the **Signature Generation** algorithm, a value $h_4 = H_4(m, ID_S, ID_V, U_S, V_S, PK_S, PK_V, h_2)$ is added, and the values of $h_2$ and $h_3$ are modified to two new values $h_2 = H_2(m, ID_S, ID_V, U_S, V_S, P_{pub})$ and $h_3 = H_3(m, ID_S, ID_V, U_S, V_S, h_2)$ respectively. The corresponding value $k_S$ is modified to

$$
k_S = h_2 d_S + h_3 t_2 + h_4 x_S \bmod q,
$$

and the signature of a message $m$ is $\sigma_S = (k_S, W_S, V_S)$.

4. The algorithms **Direct Verification** and **Public Verification** are the same as the corresponding algorithms in the original scheme. The only difference is that the verifier needs to calculate $h_2$, $h_3$ and $h_4$, and verifies $\sigma_S$ by using the following equation:

$$
k_S P = h_2(R_S + h_{1S}P_{pub}) + h_3 V_S + h_4 X_S.
$$

*Correctness*:

$$
\begin{aligned}
k_s P &= (h_2 d_s + h_3 t_2 + h_4 x_s)P \\
&= h_2(d_s P) + h_3(t_2 P) + h_4(x_s P) \\
&= h_2(R_s + h_{1S} P_{pub}) + h_3 V_S + h_4 X_S.
\end{aligned}
$$

## 5.2 Security proof

In our CLDS scheme, adequate redundant values are added to the input parameters of the three hash functions $h_2$, $h_3$ and $h_4$. By altering some public values, Type I and Type II attackers will be unable to forge valid signatures. The following theorems 1 and 2 provide the security proof for the improved CLDS scheme.

**Theorem 1**. If the ECDLP is intractable, then our improved CLDS scheme is unforgeable against Type I attacks.

**Proof**. Let $\mathcal{A}_1$ be a Type I attacker. If $\mathcal{A}_1$ can forge a legal signature of the improved scheme, then we construct an algorithm $\mathcal{C}_1$ that can solve the ECDLP. Note that $\mathcal{C}_1$ serves as the challenger in Game 1 (as stated in Section 2.2). Suppose that $\mathcal{C}_1$ obtains an ECDLP instance $(P, \alpha P)$. To calculate the unknown $\alpha \in Z_q^*$ by using the forgery of $\mathcal{A}_1$, $\mathcal{C}_1$ plays the following interactive game with $\mathcal{A}_1$.

- **Initialization Phase**: $\mathcal{C}_1$ generates system parameters *params* = $\{G, q, P, H_1, H_2, H_3, P_{pub}\}$ by executing the **Setup** algorithm, where $P_{pub} = \alpha P$. The assignment of $P_{pub}$ indicates that the master secret key is $\alpha$, but $\mathcal{C}_1$ does not know $\alpha$. Then, $\mathcal{C}_1$ selects a target user's identity $ID^*$, and sends *params* to $\mathcal{A}_1$.

- **Queries Phase**: $\mathcal{C}_1$ responds to $\mathcal{A}_1$'s various queries by establishing the following oracles.

  - **$H_1$ Oracle $\mathcal{O}_{H_1}$**: $\mathcal{C}_1$ creates a list $L_1$ of tuple $(ID_i, R_i, P_{pub}, h_{1i})$ whose initial value is empty. When $\mathcal{A}_1$ initiates a query $H_1(ID_i, R_i, P_{pub})$, $\mathcal{C}_1$ returns $h_{1i}$ to $\mathcal{A}_1$ if $L_1$ contains the tuple $(ID_i, R_i, P_{pub}, h_{1i})$. Otherwise, $\mathcal{C}_1$ randomly selects $h_{1i} \in Z_q^*$, adds $(ID_i, R_i, P_{pub}, h_{1i})$ to $L_1$, and returns $h_{1i}$ to $\mathcal{A}_1$.

  - **$H_2$ Oracle $\mathcal{O}_{H_2}$**: $\mathcal{C}_1$ creates a list $L_2$ of tuple $(m_i, ID_i, ID_j, U_i, V_i, P_{pub}, h_{2i})$ whose initial value is empty. When $\mathcal{A}_1$ initiates a query $H_2(m_i, ID_i, ID_j, U_i, V_i, P_{pub})$, $\mathcal{C}_1$ returns $h_{2i}$ to $\mathcal{A}_1$ if $L_2$ contains the tuple $(m_i, ID_i, ID_j, U_i, V_i, P_{pub}, h_{2i})$. Otherwise, $\mathcal{C}_1$ randomly selects $h_{2i} \in Z_q^*$, adds $(m_i, ID_i, ID_j, U_i, V_i, P_{pub}, h_{2i})$ to $L_2$, and returns $h_{2i}$ to $\mathcal{A}_1$.

  - **$H_3$ Oracle $\mathcal{O}_{H_3}$**: $\mathcal{C}_1$ creates a list $L_3$ of tuple $(m_i, ID_i, ID_j, U_i, V_i, h_{2i}, h_{3i})$ whose initial value is empty. When $\mathcal{A}_1$ initiates a query $H_3(m_i, ID_i, ID_j, U_i, V_i, h_{2i})$, $\mathcal{C}_1$ returns $h_{3i}$ to $\mathcal{A}_1$ if $L_3$ contains the tuple $(m_i, ID_i, ID_j, U_i, V_i, h_{2i}, h_{3i})$. Otherwise, $\mathcal{C}_1$ randomly selects $h_{3i} \in Z_q^*$, adds $(m_i, ID_i, ID_j, U_i, V_i, h_{2i}, h_{3i})$ to $L_3$, and returns $h_{3i}$ to $\mathcal{A}_1$.

  - **$H_4$ Oracle $\mathcal{O}_{H_4}$**: $\mathcal{C}_1$ creates a list $L_4$ of tuple $(m_i, ID_i, ID_j, U_i, V_i, PK_i, PK_j, h_{2i}, h_{4i})$ whose initial value is empty. When $\mathcal{A}_1$ initiates a query $H_4(m_i, ID_i, ID_j, U_i, V_i, PK_i, PK_j, h_{2i})$, $\mathcal{C}_1$ returns $h_{4i}$ to $\mathcal{A}_1$ if $L_4$ contains the tuple $(m_i, ID_i, ID_j, U_i, V_i, PK_i, PK_j, h_{2i}, h_{4i})$. Otherwise, $\mathcal{C}_1$ randomly selects $h_{4i} \in Z_q^*$, adds $(m_i, ID_i, ID_j, U_i, V_i, PK_i, PK_j, h_{2i}, h_{4i})$ to $L_4$, and returns $h_{4i}$ to $\mathcal{A}_1$.

  - **Reveal Partial Secret Key Oracle $\mathcal{O}_{PSK}$**: $\mathcal{C}_1$ creates a list $L_{PSK}$ of tuple $(ID_i, d_i, R_i, r_i)$ whose initial value is empty. $\mathcal{C}_1$ returns $D_i = (R_i, d_i)$ to $\mathcal{A}_1$ if $L_{PSK}$ contains the tuple $(ID_i, d_i, R_i, r_i)$. Otherwise, $\mathcal{C}_1$ executes as follows.

a. If $ID_i = ID^*$, $\mathcal{C}_1$ randomly selects $r^* \in Z_q^*$, sets $d^* = \bot$, calculates $R^* = r^* P$, adds $(ID^*, d^*, R^*, r^*)$ to $L_{PSK}$, and returns $D^* = (d^*, R^*)$ to $\mathcal{A}_1$.

b. If $ID_i \neq ID^*$, $\mathcal{C}_1$ randomly selects $d_i, h_{1i} \in Z_q^*$, calculates $R_i = d_i P - h_{1i} P_{pub}$, adds $(ID_i, R_i, P_{pub}, h_{1i})$ to $L_1$, records $(ID_i, d_i, R_i, r_i)$ in $L_{PSK}$, and returns $D_i = (R_i, d_i)$ to $\mathcal{A}_1$.

- **Create User Oracle** $\mathcal{O}_{CU}$: $\mathcal{C}_1$ creates a list $L_{USER}$ of tuple $(ID_i, x_i, X_i, SK_i, PK_i)$ whose initial value is empty. When receiving an identity $ID_i$ from $\mathcal{A}_1$, $\mathcal{C}_1$ returns $PK_i$ to $\mathcal{A}_1$ if $L_{USER}$ contains the tuple $(ID_i, x_i, X_i, SK_i, PK_i)$. Otherwise, $\mathcal{C}_1$ randomly selects $x_i \in Z_q^*$, retrieves $d_i$ and $R_i$ from the tuple $(ID_i, d_i, R_i, r_i)$ in $L_{PSK}$, calculates $X_i = x_i P$, sets $SK_i = (x_i, d_i)$ and $PK_i = (X_i, R_i)$, adds $(ID_i, x_i, X_i, SK_i, PK_i)$ to $L_{USER}$, and returns $PK_i$ to $\mathcal{A}_1$.

- **Reveal Secret Key Oracle** $\mathcal{O}_{SK}$: On receiving an identity $ID_i$ from $\mathcal{A}_1$, $\mathcal{C}_1$ finds the tuple $(ID_i, x_i, X_i, SK_i, PK_i)$ from $L_{USER}$ to recover $SK_i$, and then returns $SK_i$ to $\mathcal{A}_1$.

- **Replace Public Key Oracle** $\mathcal{O}_{RK}$: On receiving a tuple $(ID_i, x_i^*, PK_i^*)$ from $\mathcal{A}_1$, $\mathcal{C}_1$ looks for the tuple $(ID_i, x_i, X_i, SK_i, PK_i)$ from $L_{USER}$, and then replaces $x_i$ and $PK_i$ with $x_i^*$ and $PK_i^*$, respectively.

- **Sign Oracle** $\mathcal{O}_S$: When receiving two identities $(ID_i, ID_j)$ and a message $m_i$ from $\mathcal{A}_1$, $\mathcal{C}_1$ does as follows.

    a. If $ID_i \neq ID^*$, $\mathcal{C}_1$ runs the algorithm **Signature Generation** to produce a signature $\sigma_i = (k_i, W_i, V_i)$ of $m_i$, and returns it to $\mathcal{A}_1$.

    b. If $ID_i = ID^*$, $\mathcal{C}_1$ randomly selects $t_i, k_i, h_{2i}, h_{3i}, h_{4i} \in Z_q^*$, searches the tuple $(ID^*, x^*, X^*, SK^*, PK^*)$ in $L_{USER}$ to recover $PK^* = (X^*, R^*)$, looks for the tuple $(ID_j, x_j, X_j, SK_j, PK_j)$ in $L_{USER}$ to retrieve $x_j$ and $PK_j = (X_j, R_j)$, and finds the tuple $(ID^*, R^*, P_{pub}, h_{1i}^*)$ in $L_1$ to retrieve $h_{1i}^*$. Then, $\mathcal{C}_1$ calculates $W_i = t_i P$, $V_i = h_{3i}^{-1}(k_i P - h_{2i}(R^* + h_{1i}^* P_{pub}) - h_{4i} X^*)$ and $U_i = W_i - x_j V_i$, sets $\sigma_i = (k_i, W_i, V_i)$, and sends the signature $\sigma_i$ to $\mathcal{A}_1$. Finally, $\mathcal{C}_1$ records $(m_i, ID_i, ID_j, U_i, V_i, P_{pub}, h_{2i})$, $(m_i, ID_i, ID_j, U_i, V_i, h_{2i}, h_{3i})$ and $(m_i, ID_i, ID_j, U_i, V_i, PK_i, PK_j, h_{2i}, h_{4i})$ in lists $L_2$, $L_3$ and $L_4$, respectively.

- **D. Verify Oracle** $\mathcal{O}_{DV}$: When receiving two identities $(ID_i, ID_j)$, a message $m_i$ and a signature $\sigma_i$ from $\mathcal{A}_1$, $\mathcal{C}_1$ runs the algorithm **Direct Verification** and returns the corresponding output to $\mathcal{A}_1$.

- **P. Verify Oracle** $\mathcal{O}_{PV}$: When receiving two identities $(ID_i, ID_j)$, a message $m_i$ and a signature $\sigma_i$ from $\mathcal{A}_1$, $\mathcal{C}_1$ runs the algorithm **Public Verification** and returns the corresponding output to $\mathcal{A}_1$.

- **Forgery Phase**: Finally, $\mathcal{A}_1$ outputs a legal signature $\sigma_i^* = (k_i^*, W_i^*, V_i^*)$ for a message $m_i^*$ under the target user's identity $ID^*$ and public key $PK^*$. According to Forking Lemma [31], $\mathcal{C}_1$ uses the same random tape to replay $\mathcal{A}_1$ and assigns a different value to the hash function $H_2$, then $\mathcal{A}_1$ generates another signature $\sigma_i' = (k_i', W_i', V_i')$ of $m_i^*$ that satisfies $h_{2i}' \neq h_{2i}^*$, $h_{3i}' = h_{3i}^*$ and $h_{4i}' = h_{4i}^*$. Since $\sigma_i^*$ and $\sigma_i'$ are two legal signatures, we get the following equations.

$$k_i^* P = h_{2i}^*(R^* + h_{1i}^* P_{pub}) + h_{3i}^* V_i^* + h_{4i}^* X^* \tag{1}$$

$$k_i' P = h_{2i}'(R^* + h_{1i}^* P_{pub}) + h_{3i}' V_i^* + h_{4i}' X^* \tag{2}$$

From Eqs (1) and (2), we have

$$k_i^* = h_{2i}^*(r^* + h_{1i}^*\alpha) + h_{3i}^*t_i^* + h_{4i}^*x^* \bmod q, \tag{3}$$

$$k_i' = h_{2i}'(r^* + h_{1i}^*\alpha) + h_{3i}'t_i^* + h_{4i}'x^* \bmod q. \tag{4}$$

Thus, we use Eqs (3) and (4) to calculate

$$\alpha = (h_{2i}^* - h_{2i}')^{-1}(h_{1i}^*)^{-1}(k_i^* - k_i') - (h_{1i}^*)^{-1}(r^*)^{-1} \bmod q. \tag{5}$$

From Eq (5), we can get the solution $\alpha$ of the given ECDLP instance. However, ECDLP is a difficult problem that cannot be solved in polynomial time, so it can be inferred that the above forgery attack initiated by $\mathcal{A}_1$ is not feasible. Therefore, our improved CLDS scheme is unforgeable against the Type I attacker.

**Theorem 2**. If the ECDLP is intractable, then our improved CLDS scheme is unforgeable against Type II attacks.

**Proof**. Let $\mathcal{A}_2$ be a Type II attacker. If $\mathcal{A}_2$ can forge a legal signature of the improved scheme, then we construct an algorithm $\mathcal{C}_2$ that can solve the ECDLP. Note that $\mathcal{C}_2$ acts as the challenger in Game 2 (as defined in Section 2.2). Suppose that $\mathcal{C}_2$ obtains an ECDLP instance $(P, \alpha P)$. To calculate the unknown $\alpha \in Z_q^*$ by using the forgery of $\mathcal{A}_2$, $\mathcal{C}_2$ plays the following interactive game with $\mathcal{A}_2$.

- **Initialization Phase**: $\mathcal{A}_2$ randomly selects $s \in Z_q^*$, calculates $P_{pub} = sP$, and sets $s$ as the master secret key. Then, $\mathcal{A}_2$ executes the **Setup** algorithm to produce system parameters *params*. Finally, $\mathcal{A}_2$ sends *params* and $s$ to $\mathcal{C}_2$. Let $ID^*$ is the identity of the target user selected by $\mathcal{C}_2$.

- **Queries Phase**: $\mathcal{C}_2$ responds to $\mathcal{A}_2$'s various queries by establishing the following oracles.

  - The oracles $\mathcal{O}_{H_1}$, $\mathcal{O}_{H_2}$, $\mathcal{O}_{H_3}$ and $\mathcal{O}_{H_4}$ are the same as in Theorem 1.

  - **Create User Oracle** $\mathcal{O}_{CU}$: $\mathcal{C}_2$ creates a list $L_{USER}$ of tuple $(ID_i, d_i, R_i, x_i, X_i, SK_i, PK_i)$ whose initial value is empty. When receiving an identity $ID_i$ from $\mathcal{A}_2$, $\mathcal{C}_2$ returns $PK_i$ to $\mathcal{A}_2$ if $L_{USER}$ contains the tuple $(ID_i, d_i, R_i, x_i, X_i, SK_i, PK_i)$. Otherwise, $\mathcal{C}_2$ runs the algorithm **Partial Secret Key Gen** to create a partial private key $D_i = (R_i, d_i)$, and then performs the following operations.

    a. If $ID_i = ID^*$, $\mathcal{C}_2$ sets $x^* = \perp$ and $X^* = \alpha P$. Then, $\mathcal{C}_2$ sets $SK^* = (x^*, d^* = d_i)$ and $PK^* = (X^*, R^* = R_i)$. Next, $\mathcal{C}_2$ adds $(ID^*, d^*, R^*, x^*, X^*, SK^*, PK^*)$ to $L_{USER}$, and returns $PK^*$ to $\mathcal{A}_2$.

    b. If $ID_i \neq ID^*$, $\mathcal{C}_2$ randomly selects $x_i \in Z_q^*$, calculates $X_i = x_i P$, sets $SK_i = (x_i, d_i)$ and $PK_i = (X_i, R_i)$, adds $(ID_i, d_i, R_i, x_i, X_i, SK_i, PK_i)$ to $L_{USER}$, and returns $PK_i$ to $\mathcal{A}_2$.

  - **Reveal Secret Key Oracle** $\mathcal{O}_{SK}$: On receiving an identity $ID_i$ from $\mathcal{A}_2$, $\mathcal{C}_2$ finds the tuple $(ID_i, d_i, R_i, x_i, X_i, SK_i, PK_i)$ from $L_{USER}$ to recover $SK_i$, and then returns $SK_i$ to $\mathcal{A}_2$.

  - **Sign Oracle** $\mathcal{O}_S$: $\mathcal{C}_2$ creates a list $L_{Sig}$ of tuple $(m_i, ID_i, ID^*, V_i, W_i, t_{2i})$ whose initial value is empty. When receiving two identities $(ID_i, ID_j)$ and a message $m_i$ from $\mathcal{A}_2$, $\mathcal{C}_2$ does as follows.

    a. If $ID_i \neq ID^*$ and $ID_j \neq ID^*$, $\mathcal{C}_2$ runs the algorithm **Signature Generation** to produce a signature $\sigma_i = (k_i, W_i, V_i)$ of $m_i$.

b. If $ID_i \neq ID^*$ and $ID_j = ID^*$, $\mathcal{C}_2$ randomly selects $t_{1i}, t_{2i} \in Z_q^*$ and calculates $U_i = t_{1i} P$, $V_i = t_{2i} P$ and $W_i = U_i - t_{2i} X^*$. Then, $\mathcal{C}_2$ runs the algorithm **Signature Generation** to generate a signature $\sigma_i = (k_i, W_i, V_i)$ of $m_i$. Finally, $\mathcal{C}_2$ records $(m_i, ID_i, ID^*, V_i, W_i, t_{2i})$ in $L_{Sig}$.

c. If $ID_i = ID^*$ and $ID_j \neq ID^*$, $\mathcal{C}_2$ randomly selects $t_i, k_i, h_{2i}, h_{3i}, h_{4i} \in Z_q^*$, searches the tuple $(ID^*, d^*, R^*, x^*, X^*, SK^*, PK^*)$ in $L_{USER}$ to recover $PK^* = (X^*, R^*)$, looks for the tuple $(ID_j, d_j, R_j, x_j, X_j, SK_j, PK_j)$ in $L_{USER}$ to retrieve $x_j$ and $PK_j = (X_j, R_j)$, and finds the tuple $(ID^*, R^*, P_{pub}, h_{1i}^*)$ in $L_1$ to retrieve $h_{1i}^*$. Then, $\mathcal{C}_2$ calculates $W_i = t_i P$, $V_i = h_{3i}^{-1}(k_i P - h_{2i}(R^* + h_{1i}^* P_{pub}) - h_{4i} X^*)$ and $U_i = W_i - x_j V_i$, and sets $\sigma_i = (k_i, W_i, V_i)$.

d. If $ID_i = ID^*$ and $ID_j = ID^*$, $\mathcal{C}_2$ randomly selects $t_i, t_{2i}, k_i, h_{2i}, h_{3i}, h_{4i} \in Z_q^*$, searches the tuple $(ID^*, d^*, R^*, x^*, X^*, SK^*, PK^*)$ in $L_{USER}$ to recover $PK^* = (X^*, R^*)$, and finds the tuple $(ID^*, R^*, P_{pub}, h_{1i}^*)$ in $L_1$ to retrieve $h_{1i}^*$. Then, $\mathcal{C}_2$ calculates $W_i = t_i P$, $V_i = h_{3i}^{-1}(k_i P - h_{2i}(R^* + h_{1i}^* P_{pub}) - h_{4i} X^*)$ and $U_i = W_i - t_{2i} X^*$, and sets $\sigma_i = (k_i, W_i, V_i)$. Finally, $\mathcal{C}_2$ records $(m_i, ID^*, ID^*, V_i, W_i, t_{2i})$ in $L_{Sig}$.
$\mathcal{C}_2$ sends the final signature $\sigma_i$ to $\mathcal{A}_2$. Moreover, $\mathcal{C}_2$ records $(m_i, ID_i, ID_j, U_i, V_i, P_{pub}, h_{2i})$, $(m_i, ID_i, ID_j, U_i, V_i, h_{2i}, h_{3i})$ and $(m_i, ID_i, ID_j, U_i, V_i, PK_i, PK_j, h_{2i}, h_{4i})$ in lists $L_2, L_3$ and $L_4$, respectively.

- **D. Verify Oracle** $\mathcal{O}_{DV}$: When receiving two identities $(ID_i, ID_j)$, a message $m_i$ and a signature $\sigma_i$ from $\mathcal{A}_2$, $\mathcal{C}_2$ determines whether $ID_j = ID^*$. If it holds, $\mathcal{C}_2$ finds the tuple $(m_i, ID_i, ID^*, V_i, W_i, t_{2i})$ from $L_{Sig}$ to extract $t_{2i}$ and calculates $Y_i = U_i = W_i - t_{2i} X^*$. Then, $\mathcal{C}_2$ runs the algorithm **Direct Verification** and returns the corresponding output to $\mathcal{A}_2$.

- **P. Verify Oracle** $\mathcal{O}_{PV}$: When receiving two identities $(ID_i, ID_j)$, a message $m_i$ and a signature $\sigma_i$ from $\mathcal{A}_2$, $\mathcal{C}_2$ determines whether $ID_j = ID^*$. If it holds, $\mathcal{C}_2$ finds the tuple $(m_i, ID_i, ID^*, V_i, W_i, t_{2i})$ from $L_{Sig}$ to extract $t_{2i}$ and calculates $Aid_i = Y_i = U_i = W_i - t_{2i} X^*$. Then, $\mathcal{C}_2$ runs the algorithm **Public Verification** and returns the corresponding output to $\mathcal{A}_2$.

- **Forgery Phase**: Finally, $\mathcal{A}_2$ outputs a legal signature $\sigma_i^* = (k_i^*, W_i^*, V_i^*)$ for a message $m_i^*$ under the target user's identity $ID^*$ and public key $PK^*$. According to Forking Lemma [31], $\mathcal{C}_2$ uses the same random tape to replay $\mathcal{A}_2$ and assigns a different value to the hash function $H_4$, then $\mathcal{A}_2$ generates another signature $\sigma_i' = (k_i', W_i', V_i')$ of $m_i^*$ that satisfies $h_{4i}' \neq h_{4i}^*$, $h_{2i}' = h_{2i}^*$. Since $\sigma_i^*$ and $\sigma_i'$ are two legal signatures, we get the following equations.

$$k_i^* P = h_{2i}^*(R^* + h_{1i}^* P_{pub}) + h_{3i}^* V_i^* + h_{4i}^* X^* \tag{6}$$

$$k_i' P = h_{2i}^*(R^* + h_{1i}^* P_{pub}) + h_{3i}^* V_i^* + h_{4i}' X^* \tag{7}$$

From Eqs (6) and (7), we have

$$k_i^* = h_{2i}^*(r^* + h_{1i}^* s) + h_{3i}^* t_i^* + h_{4i}^* \alpha \bmod q, \tag{8}$$

$$k_i' = h_{2i}^*(r^* + h_{1i}^* s) + h_{3i}^* t_i^* + h_{4i}' \alpha \bmod q. \tag{9}$$

Thus, we utilize Eqs (8) and (9) to calculate

$$\alpha = (h_{4i}^* - h_{4i}')^{-1}(k_i^* - k_i') \bmod q. \tag{10}$$

From Eq (10), we can obtain the solution $\alpha$ of the given ECDLP instance. Since ECDLP is a hard mathematical problem, the aforementioned forgery attack initiated by $\mathcal{A}_2$ is not feasible. Therefore, our improved CLDS scheme is unforgeable against the Type II attacker.

## 5.3 Performance analysis

We evaluate the computational and communication costs of the enhanced CLDS scheme and similar schemes. The execution time of various cryptographic operations is given in [32], and a summary of all operations is shown in S1 Table. In the following performance comparisons, we don't consider modular multiplication and modular addition in $Z_q^*$ because their computational overhead is so negligible.

In Azees et al.'s scheme [4] and Ahamed et al.'s scheme [7], the computational cost of calculating a signature is $T_{SM} + T_H$ = 0.167001 ms, whereas the computational cost of verifying a signature is $2T_P + T_{SM} + T_{Add} + T_H$ = 9.050491 ms. Hence, the total computational overhead is 9.217492 ms. These two schemes [4, 7] have high signature generation efficiency. Unfortunately, none of them are CLDS schemes.

In Wan et al.'s CLDS scheme [26], the computational overhead of the algorithm **Signature Generation** is $5T_{SM} + T_P + 2T_{MTP} + 2T_H + T_{Add}$ = 5.557464 ms, the computational overhead of the algorithm **Direct Verification** is $3T_P + 2T_{MTP} + 2T_H$ = 13.612061 ms, whereas the computational overhead of the algorithm **Public Verification** is $2T_P + T_{MTP} + 2T_H$ = 9.028336 ms. Hence, the total computational overhead is 28.197861 ms.

In Gayathri et al.'s CLDS scheme [29], the computational overhead of the algorithm **Signature Generation** is $3T_{SM} + 2T_H + T_{Add}$ = 0.500623 ms, the computational overhead of the algorithm **Direct Verification** is $5T_{SM} + 3T_H + 4T_{Add}$ = 0.837053 ms, whereas the computational overhead of the algorithm **Public Verification** is $5T_{SM} + 3T_H + 3T_{Add}$ = 0.835649 ms. Hence, the total computational overhead is 2.173325 ms.

In our CLDS scheme, the computational overhead of the algorithm **Signature Generation** is $3T_{SM} + 3T_H + T_{Add}$ = 0.502407 ms, the computational overhead of the algorithm **Direct Verification** is $6T_{SM} + 4T_H + 4T_{Add}$ = 1.004054 ms, whereas the computational overhead of the algorithm **Public Verification** is $6T_{SM} + 4T_H + 3T_{Add}$ = 1.002650 ms. Hence, the total computational overhead is 2.509111 ms.

The comparison results of the computational overhead between our CLDS scheme and the other CLDS schemes are shown in S1 Fig. As shown in S1 Fig, the computational cost of our scheme is less than that of Wan et al.'s scheme [26], and is almost equal to that of Gayathri et al.'s scheme [29]. However, we demonstrate in Section 4 that Gayathri et al.'s scheme [29] is insecure, and that our enhanced CLDS scheme can withstand Type I and II forgery attacks.

Similar to the evaluation of communication overhead in [32], a bilinear pairing $e$: $G_1 \times G_1 \rightarrow G_2$ is selected, where the length of an element in $G_1$ is 1024 bits. In the elliptic curve cryptosystem, the length of the prime number $q$ is 160 bits, while the length of an element in the group $G$ is 320 bits. The signature length of Azees et al.'s scheme [4] is $|G_1|$ = 1024 bits = 128 bytes, the signature length of Ahamed et al.'s scheme [7] is $|G_1|$ = 128 bytes, the signature length of Wan et al.'s scheme [26] is $4|G_1|$ = 4 × 1024 = 4096 bits = 512 bytes, the signature length of Gayathri et al.'s scheme [29] is $2|G| + |q|$ = 2 × 320 + 160 = 800 bits = 100 bytes, whereas the signature length of our scheme is also $2|G| + |q|$ = 100 bytes. Obviously, our CLDS scheme has shorter signature length than other schemes [4, 7, 26]. The comparison results of the communication overhead of the three CLDS schemes are given in S2 Fig. Therefore, our CLDS scheme has better performance and higher security.

## 6 Conclusions

Directed signature can ensure data security as well as the signer's identity privacy, so it is very suitable for practical applications such as medical records and tax information. Gayathri et al. [29] devised and proved the first pairing-free CLDS scheme. However, in this article, we analyze the security of their CLDS scheme and find that their scheme is insecure against Type I and Type II attackers. Therefore, their CLDS scheme suffers from significant security flaws. To address the security issues, we give an enhanced CLDS scheme. The results of comparison with related schemes show that our scheme has higher security while keeping the original scheme's performance, so it is more suitable for practical scenarios.

## Supporting information

**S1 Data.**
(XLSX)

**S1 Table. The execution time of cryptographic operations.**
(PDF)

**S1 Fig. Comparison of the computational cost of CLDS schemes.**
(PDF)

**S2 Fig. Comparison of communication overhead of CLDS schemes.**
(PDF)

## Author Contributions

**Conceptualization:** Kaiqin Yang.

**Data curation:** Kaiqin Yang.

**Formal analysis:** Kaiqin Yang.

**Funding acquisition:** Kaiqin Yang.

**Investigation:** Kaiqin Yang.

**Methodology:** Kaiqin Yang.

**Project administration:** Kaiqin Yang.

**Resources:** Kaiqin Yang.

**Software:** Kaiqin Yang.

**Supervision:** Kaiqin Yang.

**Validation:** Kaiqin Yang.

**Visualization:** Kaiqin Yang.

**Writing – original draft:** Kaiqin Yang.

**Writing – review & editing:** Kaiqin Yang.

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
