## [Decision Letter · Decision Letter 0]

4 Jan 2022

PONE-D-21-34945An enhanced pairing-free certificateless directed signature schemePLOS ONE

Dear Dr. Yang,

Thank you for submitting your manuscript to PLOS ONE. After careful consideration, we feel that it has merit but does not fully meet PLOS ONE’s publication criteria as it currently stands. Therefore, we invite you to submit a revised version of the manuscript that addresses the points raised during the review process.

We look forward to receiving your revised manuscript.

Kind regards,

Pandi Vijayakumar, Ph.D

Academic Editor

PLOS ONE

Journal Requirements:

Additional Editor Comments:

Based on the comments of the reviewers, I recommend major revision for this paper.

Reviewers' comments:

Reviewer's Responses to Questions

**Comments to the Author**

1. Is the manuscript technically sound, and do the data support the conclusions?

Reviewer #1: Yes

Reviewer #2: Yes

2. Has the statistical analysis been performed appropriately and rigorously? 

Reviewer #1: Yes

Reviewer #2: Yes

3. Have the authors made all data underlying the findings in their manuscript fully available?

Reviewer #1: Yes

Reviewer #2: Yes

4. Is the manuscript presented in an intelligible fashion and written in standard English?

Reviewer #1: Yes

Reviewer #2: Yes

5. Review Comments to the Author

Reviewer #1: The author is required to analyse the following security related papers in the introduction section. Moreover, the computational cost of the proposed work should be compared with the following papers

1EAAP: Efficient anonymous authentication with conditional privacy-preserving scheme for vehicular ad hoc networks

Dual authentication and key management techniques for secure data transmission in vehicular ad hoc networks

An efficient anonymous mutual authentication technique for providing secure communication in mobile cloud computing for smart city applications

An efficient anonymous authentication and confidentiality preservation schemes for secure communications in wireless body area networks

EMBA: An efficient anonymous mutual and batch authentication schemes for vanets

BBAAS: Blockchain-Based Anonymous Authentication Scheme for Providing Secure Communication in VANETs

Reviewer #2: The author has made a genuine attempt to analyze the security of the previously proposed work on certificate less directed signature by Gayathri et al., using the Type I and Type II attacks. With necessary supporting proof for the previous work being susceptible to the attacks, the author has also proposed an enhanced version of the previous work which ensures the needed security. The author has introduced necessary redundant values to the input parameters of the hash functions to avoid the attacks. The author has given enough security analysis and also, the performance analysis suggests that, the work is novel and is suitable for publication in our journal.

But, I kindly request the author to give the following relevant works as citations in the proposed work to add value to its significance.

(i) Efficient NTRU Lattice-Based Certificateless Signature Scheme for Medical Cyber-Physical Systems

(ii) A practical group blind signature scheme for privacy protection in smart grid

(iii) Efficient and Secure Anonymous Authentication With Location Privacy for IoT-Based WBANs

6. PLOS authors have the option to publish the peer review history of their article (what does this mean?). If published, this will include your full peer review and any attached files.

Reviewer #1: No

Reviewer #2: No

---

## [Author Response · Author response to Decision Letter 0]

20 Jan 2022

We have uploaded a rebuttal letter in response to every point raised by the academic editor and reviewers, labeled 'Response to Reviewers'.

---

## [Decision Letter · Decision Letter 1]

31 Jan 2022

An enhanced pairing-free certificateless directed signature scheme

PONE-D-21-34945R1

Dear Dr. Yang,

We’re pleased to inform you that your manuscript has been judged scientifically suitable for publication and will be formally accepted for publication once it meets all outstanding technical requirements.

Kind regards,

Pandi Vijayakumar, Ph.D

Academic Editor

PLOS ONE

Additional Editor Comments (optional):

Reviewers' comments:

Reviewer's Responses to Questions

**Comments to the Author**

1. If the authors have adequately addressed your comments raised in a previous round of review and you feel that this manuscript is now acceptable for publication, you may indicate that here to bypass the “Comments to the Author” section, enter your conflict of interest statement in the “Confidential to Editor” section, and submit your "Accept" recommendation.

Reviewer #1: All comments have been addressed

Reviewer #2: All comments have been addressed

2. Is the manuscript technically sound, and do the data support the conclusions?

Reviewer #1: Yes

Reviewer #2: Yes

3. Has the statistical analysis been performed appropriately and rigorously? 

Reviewer #1: Yes

Reviewer #2: Yes

4. Have the authors made all data underlying the findings in their manuscript fully available?

Reviewer #1: Yes

Reviewer #2: Yes

5. Is the manuscript presented in an intelligible fashion and written in standard English?

Reviewer #1: Yes

Reviewer #2: Yes

6. Review Comments to the Author

Reviewer #1: (No Response)

Reviewer #2: (No Response)

7. PLOS authors have the option to publish the peer review history of their article (what does this mean?). If published, this will include your full peer review and any attached files.

Reviewer #1: No

Reviewer #2: No

---

## [Editor Report · Acceptance letter]

8 Feb 2022

PONE-D-21-34945R1 

An enhanced pairing-free certificateless directed signature scheme 

Dear Dr. Yang:

I'm pleased to inform you that your manuscript has been deemed suitable for publication in PLOS ONE. Congratulations! Your manuscript is now with our production department. 

Kind regards, 

on behalf of

Dr. Pandi Vijayakumar 

Academic Editor

PLOS ONE